# MeMo: Meaningful, Modular Controllers via Noise Injection

**Megan Tjandrasuwita**
MIT
megantj@mit.edu

**Jie Xu**
NVIDIA
jiex@nvidia.com

**Armando Solar-Lezama**
MIT
asolar@csail.mit.edu

**Wojciech Matusik**
MIT
wojciech@mit.edu

## Abstract

Robots are often built from standardized assemblies, (e.g. arms, legs, or fingers), but each robot must be trained from scratch to control all the actuators of all the parts together. In this paper we demonstrate a new approach that takes a single robot and its controller as input and produces a set of modular controllers for each of these assemblies such that when a new robot is built from the same parts, its control can be quickly learned by reusing the modular controllers. We achieve this with a framework called MeMo which learns (Me)aningful, (Mo)dular controllers. Specifically, we propose a novel modularity objective to learn an appropriate division of labor among the modules. We demonstrate that this objective can be optimized simultaneously with standard behavior cloning loss via noise injection. We benchmark our framework in locomotion and grasping environments on simple to complex robot morphology transfer. We also show that the modules help in task transfer. On both structure and task transfer, MeMo achieves improved training efficiency to graph neural network and Transformer baselines.[1]

## 1   Introduction

Consider the following scenario: A roboticist is designing a robot with 6 legs, such as the one seen in the left image of Fig. 1, and has trained a standard neural network controller with deep reinforcement learning (RL) to control the actuators circled in green. However, after more testing, they realize that the design of the robot needs to be extended with another pair of legs to support the desired amount of weight. Even though the new 8 leg robot is still composed of the same standard assemblies, the roboticist is unable to reuse any part of the 6 leg robot's controller. While many works [1, 2, 3] have studied structure transfer, or transferring neural network controllers to different robot morphologies, these works take a purely data-driven approach of training a universal controller on a dataset representative of the diversity and complexity of robots seen in testing. In contrast, we desire to learn transferable controllers from only a single robot and environment, obviating the requirement for a substantial training dataset and resources to perform multi-task RL. Our experiments demonstrate that state-of-the-art approaches for transferring control to environments with incompatible state-action spaces struggle to generalize in this highly data-scarce setting.

Motivated by the above scenario, we propose a framework, MeMo, for pretraining (Me)aningful (Mo)dular controllers that enable transfer from a single robot to variants with different dimensionalities. Learning transferable modules from a single robot trained on a single task is challenging, even

---

[1]Correspondence to megantj@mit.edu. Code can be found at https://github.com/MeganTj/MeMo.

38th Conference on Neural Information Processing Systems (NeurIPS 2024).

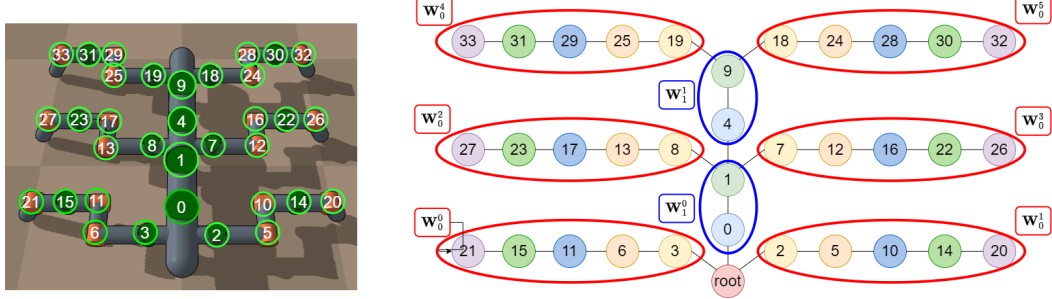

Figure 1: **Graph Structure and Neural Network Modules of the 6 Leg Centipede. Left:** The robot's joints are labeled numerically and circled. **Right:** The joints form the nodes and the links are the edges. The subset of joints that form each leg module are circled in red, while those that comprise each body module are circled in blue. Neural network modules are denoted as $\mathbf{W}_k^i$, where $k$ refers to the type, e.g. all leg modules are type 0, and $i$ denotes different instances of the same module type.

when we focus on transfer among robots with similar global morphologies. The key insight MeMo leverages is that a robot is built from assemblies of individual components, such as the leg of a walking robot or the arm of a claw robot. These assemblies are specified by a domain expert who is able to account for the constraints imposed by the robot's hardware implementation in their specification. Given this information, MeMo learns assembly-specific controllers, or modules, responsible for coordinating the individual actuators that comprise a given assembly, which are coordinated by a higher-level boss controller. As we are able to reuse the modules when transferring to a robot built from the same assemblies, the problem of learning a controller for a different morphology boils down to learning the coordination mechanics among assemblies, rather than having to coordinate at the granular level of individual joints. Returning to the 6 leg robot in Fig. 1, we see that the robot is comprised of multiple "leg" and "body" assemblies, circled in red and blue respectively in the right image. Module parameters are shared between assemblies of the same type, providing multiple training instances that help our modules generalize. After training the modules with MeMo, the "leg" and the "body" modules can then be reused to speed up the training of a different robot's controller, such as an 8 leg robot.

To achieve this improved training efficiency, a key challenge is to balance the labor between the boss controller and the modules. In one direction, to prevent the modules from becoming too robot-specialized, we introduce information asymmetry into our architecture, where the modules are limited to seeing the local observations of the actuators that belong in the module. In the other direction, controlling the assembly through the module must be simpler than controlling the assembly directly, since otherwise there is no benefit to this new architecture. This is achieved by a new modularity objective (Section 2) that forces the modules to capture as much of the coordination mechanics within a subassembly as possible, given limited local observations. In practice, we use noise injection (Section 3.1) to optimize the new objective simultaneously with standard behavior cloning loss.

To evaluate the transferability of the learned module, we apply MeMo in locomotion and grasping domains. We design two types of transfer: generalizing to more complex robot structures and to different tasks. When transferring model weights from a simpler agent, we show that MeMo significantly improves the sample efficiency of performing RL on more complex agents. We compare our framework with NerveNet, an alternative approach for one-shot structure transfer [4, 5], and MetaMorph [3], an approach for learning universal controllers. Our experiments show that MeMo either exceeds or matches NerveNet and MetaMorph's training efficiency during transfer, as the message-passing policies are prone to overfitting during pretraining.

## 2 Motivation for Modularity Objectives

Our goal is to maximize the extent to which the assembly-specific modules take responsibility for the behavior of the robot. In this section, we formalize the objectives that our training pipeline should achieve. Given an expert controller $\mathbf{F}$, one can train a modular controller, consisting of a higher-level $\mathbf{B}$(oss) module that sends a signal to each of the $\mathbf{W}$(orker) modules, to mimic $\mathbf{F}$'s behavior using the

standard behavior cloning objective. Let $\mathbf{B}$ be parameterized by $\theta$ and $\mathbf{W}$ be parameterized by $\phi$. For simplicity, this section assumes that there is a single $\mathbf{W}$ module.

**Definition 2.1. Behavior Cloning Objective.** For a system with states $s_i \in \mathcal{S}$, the modular policy whose output is $\mathbf{W}_\phi(\mathbf{B}_\theta(s_i))$ imitates the expert monolithic policy $\mathbf{F}$.

$$\text{argmin}_{\theta,\phi} \; \mathbb{E}_i \left[ \left( \mathbf{W}_\phi \left( \mathbf{B}_\theta(s_i) \right) - \mathbf{F}(s_i) \right)^2 \right] \tag{1}$$

The behavior cloning objective ensures that the composition of modules can perform the desired task, but it is not enough to ensure that the worker module is *useful*. In software engineering, a component is most useful if it can provide a narrow interface to a rich set of functionality. In the context of modularity, this is analogous to $\mathbf{W}$ giving $\mathbf{B}$ only a few degrees of freedom to control the system's outputs; otherwise, a module $\mathbf{W}$ that gives $\mathbf{B}$ full control over the actuators will leave $\mathbf{W}$ with no real responsibility for the robot's behavior. Then, when $\mathbf{W}$ is used with a new robot or for a slightly different task, $\mathbf{B}$ needs to relearn all the details of how to control the output for that new setting.

For example, consider a robot arm with 5 degrees of freedom that controls a lever in Fig. 2. An ideal worker module would take as input a signal corresponding to the desired angle of the lever and would be responsible for coordinating the signals to the five actuators to achieve the lever's desired position. This would mean that if we want to reuse $\mathbf{W}$ in controlling two arms, the new boss $\mathbf{B}'$ will only have to learn how to coordinate the two angles, and not all 10 actuators.

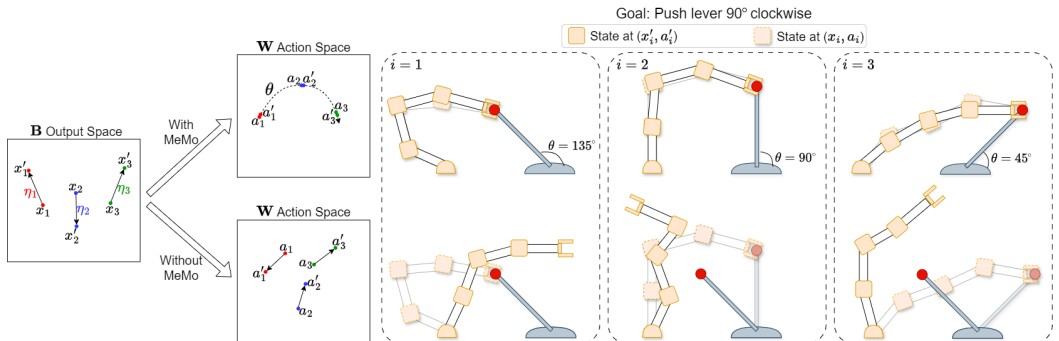

Figure 2: **Effect of Modularity Objectives**. Consider a module with 5 actuators, denoted in orange, trained to push a lever clockwise. As the state of the lever is a function of its angle $\theta$, a module trained by MeMo represents the control signals as a one-dimensional manifold with respect to $\mathbf{B}$'s signal. When noise is added to $\mathbf{B}$'s signal, the outputted actions remain on the manifold. Without MeMo, perturbations to $\mathbf{B}$'s signals cause deviations from the high reward trajectory.

In practice, though, forcing the interface between $\mathbf{B}$ and $\mathbf{W}$ to be a one dimensional vector makes the optimization problem very difficult. Instead, our approach will be to use a larger vector as the interface between the two modules, but introduce an additional optimization objective. Intuitively, the interface is effectively narrow when $\mathbf{B}$'s signal can be decomposed into a small set of dimensions that result in greater variance in $\mathbf{W}$'s output and a much larger set of dimensions that do not cause significant perturbations in $\mathbf{W}$'s output. In other words, the null space of $\mathbf{W}$'s Jacobian would be higher-dimensional, meaning that $\mathbf{W}$ has a greater tolerance for error in $\mathbf{B}$'s signal that fall in the directions of the null space. To encourage $\mathbf{W}$ to be less sensitive to perturbations in $\mathbf{B}$'s signal, we minimize the distance between $\mathbf{W}_\phi(\mathbf{B}_\theta(s_i) + \eta)$, where $\eta$ is a noise vector, and $\mathbf{W}_\phi(\mathbf{B}_\theta(s_i))$.

**Definition 2.2. Invariance to Noise Objective.** Let $\eta$ be a noise vector. The difference between the result of applying $\mathbf{W}$ on the distorted input and on the undistorted input is $D(s_i, \eta) = \mathbf{W}_\phi(\mathbf{B}_\theta(s_i) + \eta) - \mathbf{W}_\phi(\mathbf{B}_\theta(s_i))$. As a new distortion to $\mathbf{B}_\theta(s_i)$ is added on each epoch, we average the difference over the added noise.

$$\text{argmin}_{\theta,\phi} \; \mathbb{E}_\eta \left[ \mathbb{E}_i \left[ D(s_i, \eta)^2 \right] \right] \tag{2}$$

In practice, we sample $\eta$ from a Gaussian distribution with $\mathbb{E}[\eta] = 0$ and $\mathbb{E}[\eta^T \eta] = \sigma^2 \mathbf{I}$, where $\sigma = 1.0$. In Section 5.3, we demonstrate that our invariance to noise objective is the critical component in our framework that yields positive transfer benefits. In Section 5.4, we show that optimizing the noise invariance objective reduces the effective dimensionality of $\mathbf{B}$'s signal.

# 3 Method

We describe our approach MeMo, an algorithm for learning reusable control modules. In Section 3.1, we show that the modularity objectives can be optimized with noise injection. In Section 3.2, we extend our formulation to systems with more than one module and detail our training pipeline.

## 3.1 Objective

We propose to optimize both modularity objectives simultaneously with noise injection.

**Definition 3.1. Noise Injection Objective.** Here, $\eta$ can be viewed as "injected noise."

$$\text{argmin}_{\theta,\phi} \; \mathbb{E}_\eta \left[ \mathbb{E}_i \left[ (\mathbf{W}_\phi(\mathbf{B}_\theta(s_i) + \eta) - \mathbf{F}(s_i))^2 \right] \right] \tag{3}$$

The noise injection loss can be decomposed as follows:

$$\mathcal{L} = \mathbb{E}_i \left[ (\mathbf{W}_\phi(\mathbf{B}_\theta(s_i)) - \mathbf{F}(s_i))^2 \right] + \mathbb{E}_\eta \left[ \mathbb{E}_i \left[ D(s_i, \eta)^2 \right] \right]$$
$$+ \mathbb{E}_i \left[ 2(\mathbf{W}_\phi(\mathbf{B}_\theta(s_i)) - \mathbf{F}(s_i)) \mathbb{E}_\eta \left[ D(s_i, \eta) \right] \right] \tag{4}$$

See Appendix A.4 for the derivation of the decomposition. Without the last term, which we call the product term, noise injection is equivalent to the sum of Eq. 1 and 2. Analyzing the product term further, by the Mean Value Theorem, $D(s_i, \eta) = \mathbf{W}_\phi(\mathbf{B}_\theta(s_i) + \eta) - \mathbf{W}_\phi(\mathbf{B}_\theta(s_i)) = \mathbf{J_W}(z)^\intercal \eta$ for $z \in L$, where $\mathbf{J_W}$ denotes the Jacobian of $\mathbf{W}$ with respect to $\mathbf{B}$'s output and $L$ is the line segment with $\mathbf{B}_\theta(s_i)$ and $\mathbf{B}_\theta(s_i) + \eta$ as endpoints. Applying the expectation over the noise:

$$\mathbb{E}_\eta \left[ D(s_i, \eta) \right] = \mathbb{E}_\eta \left[ \mathbf{J_W}(z)^\intercal \eta \right] \tag{5}$$

Note that $z$ depends on the value of $\eta$, so it cannot be pulled out of the expectation. However, in practice, we expect that $\mathbf{J_W}(z)^\intercal \approx \mathbf{J_W}(\mathbf{B}_\theta(s_i))^\intercal$. This implies that $\mathbb{E}_\eta \left[ D(s_i, \eta) \right] \approx \mathbf{J_W}(\mathbf{B}_\theta(s_i))^\intercal \mathbb{E}_\eta \left[ \eta \right] = 0$, making the product term negligible. Empirically, in Fig. 3, we show that this product term indeed becomes much smaller than the sum of the two modularity objectives as training proceeds.

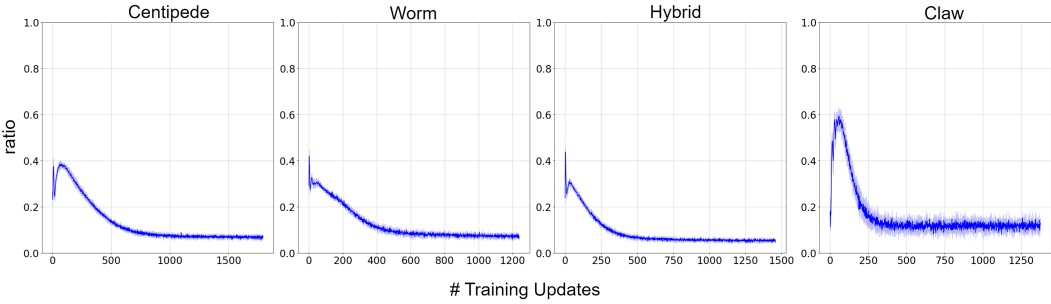

Figure 3: **Noise Injection Error.** Over the course of training, we compute ratio $= |\mathcal{L}_p|/(\mathcal{L}_1 + \mathcal{L}_2)$ where $|\mathcal{L}_p|$ is the magnitude of the mean product term over the minibatch and $\mathcal{L}_1$ and $\mathcal{L}_2$ are the mean behavior cloning and invariance to noise losses. We compute training statistics over 5 runs and indicate standard deviation by shaded areas. **(Left)-(Right):** For all starting morphologies, the modularity objectives dominate the loss as the ratio is less than 1 for all updates.

## 3.2 Modular Architecture and Training Pipeline

**Modular Architecture.** Although thus far we have only a single module $\mathbf{W}$, a robot is often comprised of multiple modules controlling physical assemblies that are common among different morphologies. Formally, we assume that we are given a partitioning $\mathcal{P}$ of an agent's joints $j_{0,\dots,N-1}$. We design a modular policy composed of a boss controller $\mathbf{B}$ that outputs intermediate signals to neural network modules that decode actions. Each element of the partition, e.g. a subset of actuators, is a module instance $i$ of type $k$, which we denote as $\mathbf{W}_k^i$. In total, there are $|\mathcal{P}|$ modules. Modules of the same type $k$ share the module parameters, yet each instance will receive a different message from $\mathbf{B}$. We detail our architecture further in Appendix A.3.

**Training Pipeline.** To train our modules, inspired by previous works that combine RL and IL [6, 7, 8], we first train $\mathbf{F}$ using RL. During the RL stage, we use proximal-policy optimization [9] to train actor-critic controllers. The critic is a MLP, whereas the actor is a standard MLP when training the expert controller and a modular architecture when transferring pretrained modules. Once $\mathbf{F}$ is trained, we train a modular policy $\pi_{\theta,\phi}(a_i \mid s_i)$ with IL via DAgger [10], with noise injected into $\mathbf{B}_\theta$'s output. At each iteration $k$ of DAgger, we sample a trajectory $\mathcal{D}_k$ from $\pi_{\theta,\phi}$. $\mathbf{F}$ provides the correct action to each $s \in \mathcal{D}_k$, and $\mathcal{D}_k$ is aggregated into the full dataset $\mathcal{D} = \{(s_i, a_i)\}$. To optimize the objective defined in Section 3.1, we minimize $\mathcal{L} = -\mathbb{E}_{s_i \sim \mathcal{D}}\left[\log \pi_{\theta,\phi}(a_i \mid s_i)\right]$. We derive a decomposition of the negative log likelihood loss with noise injection into the modularity objectives in Appendix A.5. After transferring the modules to a new structure or task, we perform RL to retrain $\mathbf{B}$ or finetune the architecture end-to-end. Our pipeline is summarized in Fig. 4. Appendix A.6 details our RL and IL hyperparameter settings.

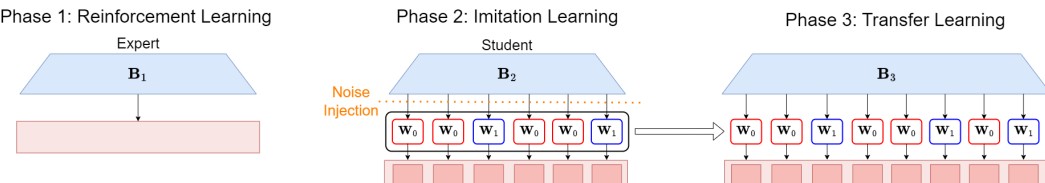

Figure 4: **Training Pipeline Overview.** In Phase 1, we first train an expert controller for the training robot using RL. In Phase 2, we pretrain modules with noise injection during imitation learning. In Phase 3, we transfer the modules to a different context and retrain the boss controller $\mathbf{B}_3$.

## 4 Related Work

Our work relates to modular controllers and structure transfer. Related works in noise injection, multi-robot RL, and hierarchical RL are discussed further in Appendix A.11.

**Modular Controllers.** Our work relates to prior works that train modular policies for robot designs. [11] learns neural network policies that are decomposed into "task-specific" and "robot-specific" modules and performs zero-shot transfer to unseen task and robot-specific module combinations. [1] coordinates modular policies shared among all actuators via message passing. [12] uses a GNN to internally coordinate between part-specific nodes with shared module parameters between nodes corresponding to the same part. [13] proposes the Dynamic Graph Network to control self-assembling agents, consisting of modules that are shared across agents.

**Structure Transfer.** In the hierarchical RL setting, [14] uses imitation learning to train policies that represent long-horizon behavior and improve sample efficiency when transferred from simple to complex agents. [15] transfers policies to robots with significantly different kinematics and morphology by defining a continuous evolution from the source to the target robot. Previous works use message-passing policy architectures to generalize across morphologies [1, 4, 5]. In the multi-task setting [2] proposes Transformers as policy representations that remove the need for multi-hop communication. [3] scales Transformer-based policies to larger and more diverse datasets of robots.

## 5 Experiments

With our experiments, we seek to answer four questions. 1) Do the modules produced by MeMo generalize when transferred to different robot morphologies and tasks? 2) When pretraining modular controllers with imitation learning, does the Gaussian noise injection help? 3) In the pretraining phase, why do we use imitation learning rather than injecting noise in reinforcement learning? 4) How does our modularity objective yield better representations of the actuator space? We answer question 1) in Sections 5.1 and 5.2, 2) and 3) in Section 5.3, and 4) in Section 5.4.

### 5.1 Transfer Learning

We benchmark our framework on two types of transfer: structure and task transfer. While our framework is designed primarily for structure transfer, we use task transfer experiments as an additional means of evaluating the quality of the learned representations. For the locomotion

experiments, we perform experiments on the tasks introduced in RoboGrammar [16] with training statistics computed as the average reward across 3 runs, with standard deviations indicated by shaded areas. For the grasping domain, we construct object-grasping tasks using the DiffRedMax simulator [17] and compute training statistics as the average reward across 5 runs. Additional details on the reward functions used are in Appendix A.9. We visualize train and test robot morphologies for structure transfer in Fig. 5 and the train and test tasks for task transfer in Fig. 18.

**Locomotion.** We design three structure transfer tasks in the locomotion domain, in which the goal is to move as far as possible while maintaining the robot's initial orientation. The starting morphologies are the 6 leg centipede robot, the 6 leg worm robot, and the 6 leg hybrid. The 6 to 12 leg centipede transfer demonstrates scalability to transfer robots with many more modules than seen in training. The 6 to 10 leg worm shows that MeMo generalizes with only 1-2 instances of the same module seen in training. The 6 and 10 leg hybrid robots involve three types of modules, demonstrating scalability to more complex training robots. For task transfer, we transfer policy weights pretrained on a 6 leg centipede locomoting over the Frozen Terrain to three terrains that feature obstacles or climbing.

**Grasping.** In grasping, the goal is to grasp and lift an object as high as possible. We design a grasping robot consisting of an arm that lifts a claw grasping a cube. The structure transfer is from a 4 finger to a 5 finger claw. For task transfer, we transfer policies trained to control the 4 finger claw grasping a cube to the same robot grasping a sphere of similar size and weight.

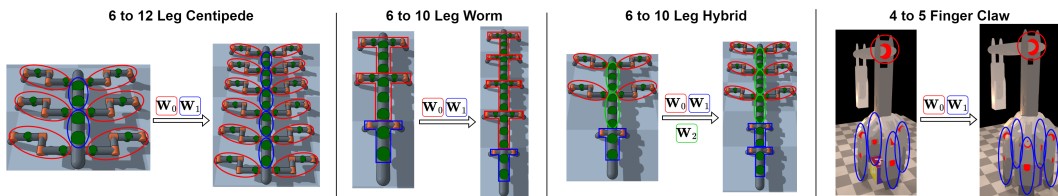

Figure 5: **Structure Transfer Tasks. Left:** Transfer "leg" and "body" modules from a 6 to a 12 leg centipede. **Left Middle:** Transfer "body" and "head" modules from a 6 to a 10 leg worm. **Right Middle:** Transfer "leg," "head," and "body" modules from a 6 to a 10 leg hybrid. **Right:** Transfer "arm" and "finger" modules from a 4 to a 5 finger claw.

**Baselines.** We compare MeMo to MLP and modular policies trained from scratch as well as pretrained NerveNet [4, 5] and MetaMorph [3] baselines. NerveNet takes as input the underlying graph structure of the agent, where the nodes are actuators and edges are body parts. The graph structures of the train morphologies are detailed in Appendix A.13. For MetaMorph, a Transformer-based approach, we convert the global observations and local observations for each actuator to a 1D sequence of tokens. Full training details and state space descriptions are included in Appendix A.6 and A.7 respectively.

- **RL (MLP)**: For structure transfer, due to the change in the observation space, we train a 2 layer MLP policy from scratch with RL. In task transfer, we use a MLP pretrained with RL on the original task and finetune it on the test task. For a fair comparison, we use the same architecture size as the modular architecture's boss controller and replace the modules with a linear layer decoder.
- **RL (Modular)**: For structure transfer, we train the modular architecture, discussed in Section 3.2, from scratch with RL. In task transfer, we use the modular architecture pretrained with RL on the training task and finetune both the modules and the boss controller on the test task. The inclusion of this baseline allows us to isolate the effect of the modular architecture from the pretraining and noise injection components of MeMo.
- **Pretrained NerveNet-Conv**: We use the NerveNet network architecture proposed by [4], consisting of an input network $F_{in}$ for encoding observations, a message function $M$, an update network $U$, and an output network $F_{out}$ for decoding. As in [4], $F_{in}$ and $F_{out}$ are MLPs. In the convolutional [18] variant, $M$ is the identity function and $U$ is a weight matrix. During RL, we fix $F_{out}$ in a similar manner as fixing the modules in MeMo, which improves NerveNet-Conv's performance.
- **Pretrained NerveNet-Snowflake:** Snowflake [5] is a state-of-the-art approach for training GNN policies that scale to high-dimensional continuous control. Their method involves fixing parts of the NerveNet architecture to prevent overfitting during PPO. Empirically, they find that fixing $\{F_{in}, M, F_{out}\}$ results in the best performance on MuJoCo tasks. We follow the same parameter fixing as Snowflake. As in Snowflake, we parameterize $F_{in}$ and $F_{out}$ as MLPs and the update function $U$ as a GRU. We use a weight matrix for $M$.

- **Pretrained MetaMorph:** MetaMorph [3] is a Transformer-based approach for learning a universal controller over a large collection of robot morphologies. We adopt MetaMorph's Transformer architecture for the policy network, which adds learned positional embeddings before processing the input sequence with a Transformer encoder. As our domains lack exteroceptive observations, we directly decode Transformer encodings to controller outputs. The Transformer policy is finetuned during RL.

## 5.2 Results

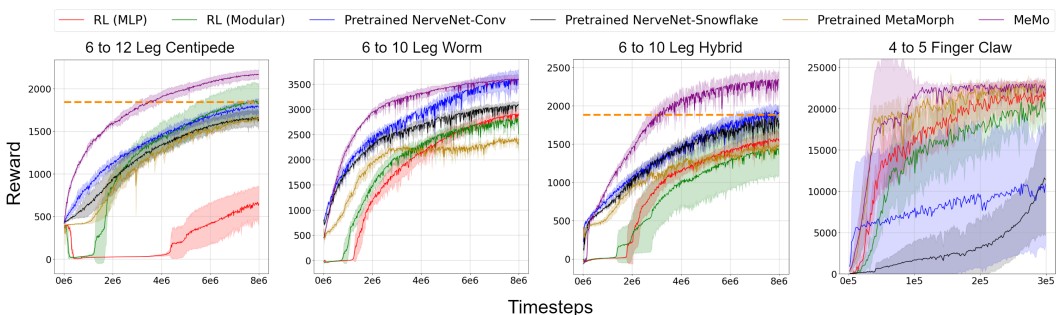

Figure 6: **Structure Transfer Results. Left:** 6 leg centipede to 12 leg centipede transfer on the Frozen Terrain. **Left Middle:** 6 leg worm to 10 leg worm transfer on the Frozen Terrain. **Right Middle:** 6 leg hybrid to 10 leg hybrid transfer on the Frozen Terrain. **Right:** 4 finger claw to 5 finger claw transfer on grasping a cube. The dashed orange line shows that the final performance of the closest baseline is achieved by MeMo within half of the total number of timesteps.

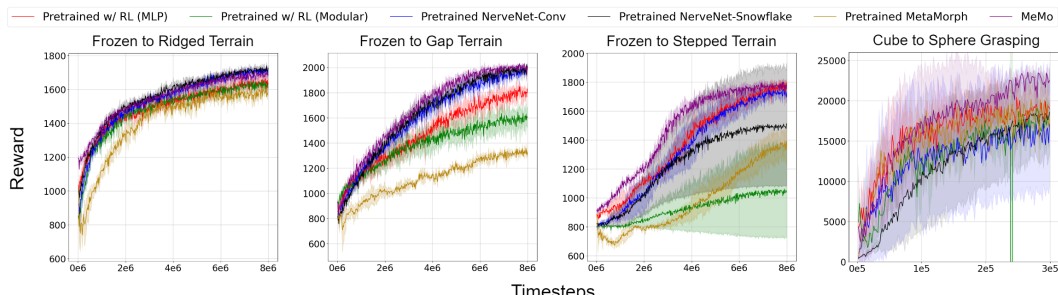

Figure 7: **Task Transfer Results. Left:** The first three plots show results on transferring from the 6 leg centipede walking over the Frozen Terrain to the same centipede walking over a terrain with ridges, a terrain with gaps, and a terrain with upward steps. **Right:** The last plot shows the transfer from a 4-finger claw grasping a cube to the same claw grasping a sphere. MeMo either has comparable training efficiency to the strongest baseline or outperforms all baselines.

The generalization ability of MeMo on structure transfer is shown in Fig. 6. On all structure transfer tasks, MeMo outperforms the message-passing baselines. On the 12 leg centipede and the 10 leg hybrid, not only is MeMo $2\times$ more sample efficient than the best baseline, but it also converges to controllers with significantly better performance than any baseline. On the 10 leg worm, MeMo outperforms all baselines in terms of training efficiency and achieves a comparable final performance as NerveNet-Conv. MeMo also outperforms all baselines on the 5 finger claw. We note that the worm transfer task is easier for GNN models, because the coordination of the shorter legs and body joints is naturally captured with multi-hop communication. MetaMorph struggles with locomotion tasks, due to the high dimensionalities of the transfer robots.

The results of MeMo on task transfer are shown in Fig. 7. As transferring from the Frozen to the Ridged, Gap, and Stepped Terrains requires the robot to overcome obstacles unseen in the Frozen Terrain, we load the pretrained boss controller and finetune MeMo end-to-end. Results (Fig. 7) show that on all test tasks, MeMo achieves improved training efficiency compared to MetaMorph and to pretrained MLP and modular architectures. MeMo achieves comparable performance on the Ridged

and Gap Terrains and outperforms the NerveNet baselines on the Stepped Terrain, which requires the robot to climb up steps whereas the training terrain is flat. MeMo also has improved training efficiency and final performance in the grasping domain when transferring from grasping a cube to a sphere. The pretrained NerveNets struggle to coordinate the arm and claw components, resulting in high variance across different random seeds.

## 5.3 Ablation Study

**Sum of Modularity Objectives.** We answer the question of why we choose to optimize the noise injection objective rather than the sum of the modularity objectives directly. We evaluate the sum of Eq. 1 and 2 between networks trained with the noise injection objective and those trained with the sum in Table 1. Using 100 sampled trajectories from the expert controller, we average the resulting sum of objectives over 1000 epochs, with different sampled noise on each epoch. Our results demonstrate that optimizing the noise injection objective converges to better solutions.

Table 1: **Sum of objectives.** On all starting morphologies, optimizing the noise injection objective results in lower loss than directly optimizing the dual loss.

| MORPHOLOGY | NOISE INJECTION | DUAL LOSS |
|---|---|---|
| CENTIPEDE | -33.518 | -33.115 |
| WORM | -39.295 | -35.896 |
| HYBRID | -30.536 | -27.849 |
| CLAW | -8.215 | 0.279 |

**Noise Injection Objective.** The key to the success of MeMo is the introduced noise injection (NI) objective which encourages proper responsibility division among the pretrained boss controller and modules, enabling the modules to improve training efficiency when reused. We conduct an ablation study to verify this technique by experimenting on a special setting, "transferring" the controller to the same robot structure and task, a 6 leg centipede traversing a Frozen Terrain. During transfer, we reuse and freeze the pretrained modules and retrain the boss controller from scratch. With the pretrained modules from MeMo, the boss controller will be retrained much more efficiently because it only needs to take partial responsibility for the control job. We compare our method to three baselines:

- **MeMo (no NI)**: We pretrain the modular architecture end-to-end without noise injection. This ablation is equivalent to MeMo without noise injection.
- **MeMo (L1):** During pretraining, we replace the injected noise with L1 regularization on **B**'s output that encourages sparsity in its signal. We weigh the regularization term by a hyperparameter $w$ and report results with the best $w$.
- **MeMo (L2):** During pretraining, we replace the injected noise with L2 regularization on **B**'s output and report results with the best weight on the regularization term.
- **MeMo (Jacobian):** As an alternative to noise injection, we penalize the norm of the module's Jacobian using the method described in [19].

In addition, we add the training curve of **RL (Modular)** as a reference. The results (Fig. 8) show that MeMo yields a significant improvement in training efficiency over all ablations.

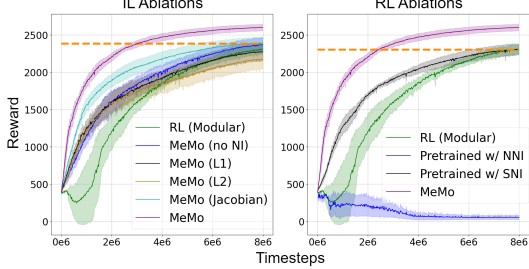

Figure 8: **Ablation Results. Left:** MeMo outperforms all other variants that are pretrained with IL. **Right:** MeMo outperforms all variants that pretrain modules with RL. In both settings, MeMo achieves the final performance of the closest baseline within half of the total number of timesteps.

**Imitation Learning.** We now answer the second question of whether imitation learning is necessary to pretrain modules with Gaussian noise injection. The results of using noise injection in reinforcement learning to pretrain modules is shown in Fig. 8. Note that we refer to IL ablations as experiments where modules are first pretrained with imitation learning, and subsequently, the boss controller is reinitialized and retrained with RL to test the improvement in sample efficiency. RL ablations involve the second RL phase, but the pretraining stage is done with RL as well. In addition to training the modular architecture from scratch, we experiment with two methods of injecting noise during RL. The first is naive noise injection (NNI), where we inject noise into **B**'s output when sampling rollouts and computing policy gradients. For the second, we adopt the Selective Noise Injection (SNI) technique proposed by [20] for applying regularization methods with stochasticity in RL. SNI stabilizes training by sampling rollouts deterministically and computing the policy gradient as a mixture of gradients from the deterministic and stochastic policies. However, even with SNI, the pretrained modules do not improve training efficiency.

## 5.4 Analysis

We examine how the noise injection objective forces the modules to learn a better representation of the actuator space. As discussed in Section 2, the trajectories produced by a successful policy often lie on a much lower-dimensional manifold than the actuator space. Each dimension of the manifold can be interpreted as an individual skill that the policy has learned. We can measure the dimensionality of the modules' mapping by looking at the Jacobian matrix of the worker modules with respect to the boss's signal. The trajectories outputted by a policy can likely be captured by a few dimensions of high variance corresponding to a small set of large singular values in addition to a much larger set of dimensions of lower variance corresponding to relatively small singular values.

We visualize this effect by 1) computing the Jacobians at the trajectory input states of a successful policy and 2) normalizing the singular values of each Jacobian by its largest singular value and plotting the resulting values in the [0, 1] range. We expect that a module that optimizes the invariance to noise objective will have only a small number of large singular values, with the rest being close to zero. Conversely, modules that do not produce a low-dimensional manifold would have more singular values of similar magnitude, resulting in the distribution's mass clustering close to 1. We verify this intuition by sampling 100 trajectories from an expert controller for the 6 leg centipede shown in Fig. 1. We compare the plots of the normalized singular values between MeMo and MeMo without noise injection in Fig. 9. Without noise injection, the majority of the values are close to 1. At the other extreme, with MeMo, the values are highly clustered to the left, implying that most singular values are much smaller than the biggest singular value. We plot the singular value distributions of additional MeMo ablations in Appendix A.12.

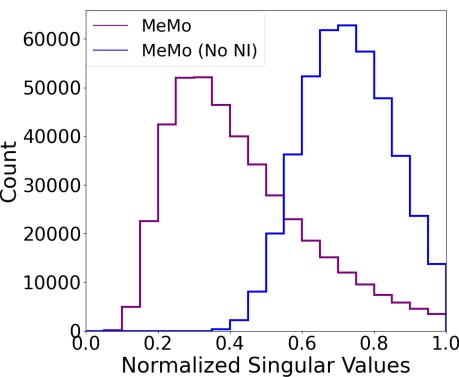

Figure 9: **Singular Value Distributions of Actuator-Boss Jacobians.** For modular architectures trained with and without the noise injection, we plot the normalized singular values of Jacobian matrices over an expert's trajectories. With noise injection, the mass of the distribution is much closer to 0, showing that the modules learn better representations of the actuator space.

## 6 Conclusion

In this paper, we propose a modular architecture for robot controllers, in which a higher-level boss controller coordinates lower-level modules that control shared physical assemblies. We train the architecture end-to-end with noise injection, which ensures that the lower-level modules do not overrely on the boss controller's signal. In locomotion and grasping environments, we demonstrate that our pretrained modules outperform both GNN and Transformer-based methods when transferring from simple to complex morphologies and to different tasks. We ablate components of MeMo and demonstrate that the entire framework is necessary to achieve these generalization benefits.

## Acknowledgments and Disclosure of Funding

MT is supported by the National Science Foundation (NSF) under Grant No. 2141064. AS is supported by the National Science Foundation (NSF) under Grant No. 1918839 and by the MIT-IBM Watson AI Lab. MT was also additionally supported by the MIT Stata Presidential Fellowship. This work greatly benefited from discussion with colleagues in the MIT Computational Design and Fabrication Group and MIT Computer-Aided Programming Group. Any opinion, findings, and conclusions or recommendations expressed in this material are those of the authors(s) and do not necessarily reflect the views of the funding entities.

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

# A Appendix

## A.1 Limitations and Future Work

We now discuss the limitations of our work and potential future directions. One limitation is that our experiments were conducted using only a single type of RL algorithm (PPO) and network architecture, MLPs with Tanh nonlinearities and orthogonal initialization. Future work would involve applying and adapting our framework to different policy optimization algorithms and architectures.

Because our approach involves imitation learning to pretrain the modules, we acknowledge that there is a potentially greater memory footprint for storing the imitation learning dataset.

While we have demonstrated the potential of MeMo to be used for task transfer, such capabilities are inherently limited as our architecture does not explicitly encode the semantics of the task. For example, transferring the modules of a 6 leg robot walking over a Frozen Terrain to the same robot traversing a terrain with obstacles requires finetuning the architecture end-to-end, which may not always yield better results due to the instability of policy updates. Combining our framework with a mechanism for representing task semantics to enable transfer to more complex tasks is a promising direction for future work. In addition, as our experiments are only in simulation, an important line of future work is applying our approach to real world tasks.

In general, we would expect our framework to face the same challenges in adapting to the real world as standard RL policies, as our problem of learning modules that generalize to different morphologies is orthogonal to learning policies that overcome the sim-to-real gap. Extending our framework to deal with challenges in sim-to-real transfer, including adapting to environmental variations, hardware inconsistencies, and discrepancies in simulated physics vs real dynamics, is an important avenue for future work.

The scope of our work is limited to robots that are incrementally different from the starting robot, due to the difficulty of generalizing from a single robot and environment. One line of future work could involve adapting our training pipeline to multirobot and multitask settings, enabling our modules to capture a broader range of robot dynamics. When the dataset of robots grows larger, it can be expensive for a domain expert to manually provide labels on how a robot is decomposed into physical assemblies. However, the problem of learning reusable components bears similarity to the problem of abstraction learning studied in the programming languages community. Recent advances [21, 22] have made abstraction learning much more computationally efficient than in the past. We also see promise in adapting these techniques, which have been developed for programs, to robots that have an underlying graph structure.

## A.2 Broader Impact

Here we discuss the broader social impact of our work, including its potential positive and negative aspects. On the positive side, our work enables us to train neural network architectures which are structured in a more interpretable manner, in that the modules correspond to physical components of the robot. In addition, we demonstrate generalization to robot structures and tasks that are greater in difficulty than the training setting. In summary, our modular approach is a step towards addressing the concern that neural networks are black-box models with highly limited generalization capabilities.

We do not see any direct negative implications stemming from our work, as experiments are solely conducted in simulated robot environments. We note that our work does not impose safety constraints on the rollouts of the agent, which is an important limitation to address for real-world use of our method.

## A.3 Further Architecture Details

As shown in Fig. 10, the modular architecture starts by executing the boss controller **B**, which takes the full observation vector $s^\tau$ as input. The full observation vector $s^\tau$ includes both global observations about the agent and local observations of each actuator. The global observations consist of the agent's global position, orientation, and velocity, while the local observations are composed of joint angle, joint velocity, and local relative position and orientation in the corresponding module's frame. Given the full observation vector $s^\tau$, $M$ outputs a latent vector $H$ of length $|\mathcal{P}| \cdot D$, where $D$ is the size of the embedding sent to each module.

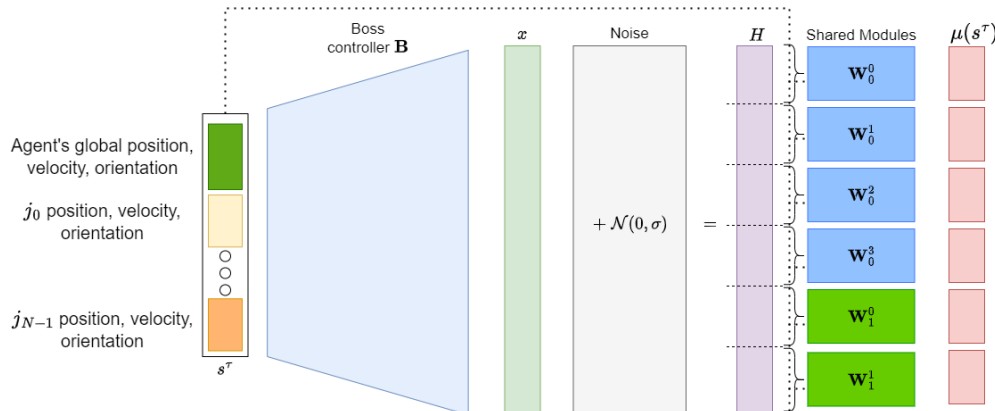

Figure 10: **Modular Architecture with Noise Injection.** Our architecture consists of a higher-level boss controller **B** that outputs a hidden embedding, $x$. During imitation learning, Gaussian noise is added to $x$ to compute $H$. $H$ is split into signals that are passed into modules that output the mean of the action distribution. The dotted lines represent that in addition to $H$, the modules also take in subsets of the full observation vector corresponding to the state of the joints within the modules.

The latent vector $H$ is then split into $|\mathcal{P}|$ segments of size $D$ and fed to modules. As shown in Fig. 11, a module itself consists of a MLP for each actuator $j_n$. Each MLP takes as input $j_n$'s local features concatenated with the module's segment of the latent vector $H$ and outputs the mean value of the action applied to $j_n$.

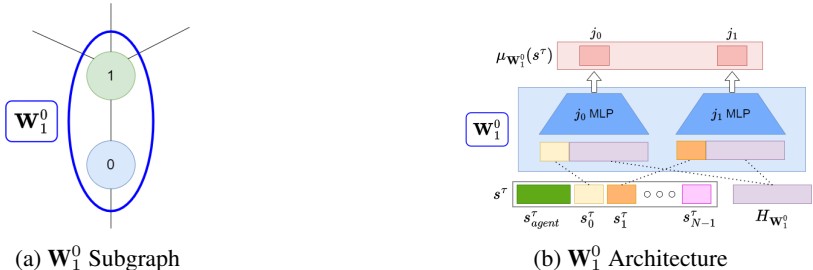

(a) $\mathbf{W}_1^0$ Subgraph        (b) $\mathbf{W}_1^0$ Architecture

Figure 11: **Module Subgraph and Architecture. Left:** Module $\mathbf{W}_1^0$ is responsible for computing the mean actions of actuators 0 and 1. **Right:** A module consists of separate networks that compute each actuator's mean action. The inputs include the local observations of the actuator concatenated with the signal sent to the module it belongs to.

## A.4  Decomposition of Noise Injection Loss

Our goal is to show that Eq. 3.1 is equivalent to the sum of Eq. 1 and 2 with a remaining product term. We can decompose the noise injection loss as follows:

$$\mathcal{L} = \mathbb{E}_\eta \left[ \mathbb{E}_i \left[ (\mathbf{W}_\phi(\mathbf{B}_\theta(s_i) + \eta) - \mathbf{F}(s_i))^2 \right] \right] \tag{6}$$

$$= \mathbb{E}_\eta \left[ \mathbb{E}_i \left[ (D(s_i, \eta) + (\mathbf{W}_\phi(\mathbf{B}_\theta(s_i)) - \mathbf{F}(s_i)))^2 \right] \right] \tag{7}$$

$$= \mathbb{E}_\eta [\mathbb{E}_i [D(s_i, \eta)^2 + 2D(s_i, \eta)(\mathbf{W}_\phi(\mathbf{B}_\theta(s_i)) - \mathbf{F}(s_i))$$
$$+ (\mathbf{W}_\phi(\mathbf{B}_\theta(s_i)) - \mathbf{F}(s_i))^2]] \tag{8}$$

$$= \mathbb{E}_i \left[ (\mathbf{W}_\phi(\mathbf{B}_\theta(s_i)) - \mathbf{F}(s_i))^2 \right] + \mathbb{E}_\eta \left[ \mathbb{E}_i \left[ D(s_i, \eta)^2 \right] \right]$$
$$+ \mathbb{E}_\eta \left[ \mathbb{E}_i \left[ 2D(s_i, \eta)(\mathbf{W}_\phi(\mathbf{B}_\theta(s_i)) - \mathbf{F}(s_i)) \right] \right] \tag{9}$$

## A.5 Negative Log Likelihood Loss

In practice, we minimize the negative log likelihood loss during imitation learning. Similar to the above derivation, show that negative log likelihood with noise injection can be written in terms of the behavior cloning and invariance to noise losses. Let $\pi$ be our modular policy, where the mean of the Gaussian distribution for each of the actions is given by $\mathbf{W}_\phi(\mathbf{B}_\theta(s_i))$, and the standard deviation is a trainable vector, $\sigma_u$. We define the behavior cloning and invariance to noise losses as negative log likelihoods in Eq. 11 and 13 respectively.

$$p_\pi(\mathbf{F}(s_i) \mid s_i) = \frac{1}{\sigma_u\sqrt{2\pi}} \exp\left(-\frac{1}{2\sigma_u^2}(\mathbf{W}_\phi(\mathbf{B}_\theta(s_i)) - \mathbf{F}(s_i))^2\right) \tag{10}$$

$$-\mathbb{E}_i\left[\log p_\pi(\mathbf{F}(s_i) \mid s_i)\right] = \mathbb{E}_i\left[\log(\sigma_u\sqrt{2\pi}) + \frac{1}{2\sigma_u^2}(\mathbf{W}_\phi(\mathbf{B}_\theta(s_i)) - \mathbf{F}(s_i))^2\right] \tag{11}$$

$$p_\pi(\mathbf{W}_\phi(\mathbf{B}_\theta(s_i)) \mid s_i, \eta) = \frac{1}{\sigma_u\sqrt{2\pi}} \exp\left(-\frac{1}{2\sigma_u^2}\left(\mathbf{W}_\phi(\mathbf{B}_\theta(s_i) + \eta) - \mathbf{W}_\phi(\mathbf{B}_\theta(s_i))\right)^2\right) \tag{12}$$

$$-\mathbb{E}_\eta\left[\mathbb{E}_i\left[\log p_\pi(\mathbf{W}_\phi(\mathbf{B}_\theta(s_i)) \mid s_i, \eta)\right]\right] = \mathbb{E}_\eta\left[\mathbb{E}_i\left[\log(\sigma_u\sqrt{2\pi}) + \frac{1}{2\sigma_u^2}(\mathbf{W}_\phi(\mathbf{B}_\theta(s_i) + \eta) - \mathbf{W}_\phi(\mathbf{B}_\theta(s_i)))^2\right]\right] \tag{13}$$

We now consider the conditional log likelihood with noise injection and show how it can be decomposed in terms of the above losses.

$$p_\pi(\mathbf{F}(s_i) \mid s_i, \eta) = \frac{1}{\sigma_u\sqrt{2\pi}} \exp\left(-\frac{1}{2\sigma_u^2}\left(\mathbf{W}_\phi(\mathbf{B}_\theta(s_i) + \eta) - \mathbf{F}(s_i)\right)^2\right) \tag{14}$$

$$-\mathbb{E}_\eta\left[\mathbb{E}_i\left[\log p_\pi(\mathbf{F}(s_i) \mid s_i, \eta)\right]\right] = \mathbb{E}_\eta\left[\mathbb{E}_i\left[\log(\sigma_u\sqrt{2\pi}) + \frac{1}{2\sigma_u^2}(\mathbf{W}_\phi(\mathbf{B}_\theta(s_i) + \eta) - \mathbf{F}(s_i))^2\right]\right] \tag{15}$$

$$= \mathbb{E}_i\left[\log(\sigma_u\sqrt{2\pi}) + \frac{1}{2\sigma_u^2}(\mathbf{W}_\phi(\mathbf{B}_\theta(s_i)) - \mathbf{F}(s_i))^2\right]$$
$$+ \mathbb{E}_\eta\left[\mathbb{E}_i\left[\frac{1}{2\sigma_u^2}\left(\mathbf{W}_\phi(\mathbf{B}_\theta(s_i) + \eta) - \mathbf{W}_\phi(\mathbf{B}_\theta(s_i))\right)^2\right]\right] + C \tag{16}$$

$$= -\mathbb{E}_i\left[\log p_\pi(\mathbf{F}(s_i) \mid s_i)\right] - \mathbb{E}_\eta\left[\mathbb{E}_i\left[\log p_\pi(\mathbf{W}_\phi(\mathbf{B}_\theta(s_i)) \mid s_i, \eta)\right]\right]$$
$$- \log(\sigma_u\sqrt{2\pi}) + C \tag{17}$$

where $C$ is the product term.

## A.6 Further Experimental Details

Let $D$ be the base hidden size of the network. As typical in PPO, we use Tanh nonlinearities and orthogonal initialization for the standard MLP and modular architectures. The standard MLP and boss controller are 2 layer neural networks. The size of the first layer is $D$ while the second layer has $L \cdot D$ hidden units, where $L$ is the number of modules. The standard MLP also has a final linear layer to decode the actions. For all policy architecture variants, the value function is defined as a 2 layer neural network with $D$ hidden units each, followed by a linear layer.

In PPO, agents iteratively sample trajectories based on the current policy and subsequently perform optimization on a surrogate objective that first-order approximates the natural gradient. The surrogate objective prevents unstable updates to the policy by clipping the probability ratio $r^\tau(\theta; \theta_{old}) = \pi_\theta(a^\tau \mid s^\tau)/\pi_{\theta_{old}}(a^\tau \mid s^\tau)$. Optimizing the clipped objective is done with the policy gradient [23].

The RL loss that all architectures optimize includes the surrogate objective, a weighted value function loss, and a weighted entropy bonus to encourage exploration:

$$L^\tau(\theta) = \mathbb{E}\left[L^\tau_{CLIP}(\theta) - c_1 L^\tau_V(\theta) + c_2 S[\pi_\theta](s^\tau)\right]$$
$$= \mathbb{E}\left[\min\left(\hat{A}^\tau r^\tau(\theta), \hat{A}^\tau \mathrm{clip}(r^\tau(\theta), 1-\epsilon, 1+\epsilon)\right)\right]$$
$$- c_1 \mathbb{E}\left[\left(V_\theta(s^\tau) - V^\tau_{targ}(s^\tau)\right)^2\right] + c_2 \mathbb{E}\left[S[\pi_\theta](s^\tau)\right] \tag{18}$$

where $\hat{A}^\tau$ is the generalized advantage estimation (GAE) [24]. $\epsilon$ is the clip value, $c_1$ is the weight on the value function, and $c_2$ is used to balance the entropy bonus.

We use Adam as the optimizer for both RL and IL. Semicolon-separated entries denote different values for the two domains: "[Locomotion Value]; [Grasping Value]". We conduct an extensive hyperparameter search and find that that the values in Table 2 yield reasonable performance.

| Parameters | Value Set |
|---|---|
| Value Loss Factor $c_1$ | 0.5 |
| Entropy Bonus Factor $c_2$ | 0 |
| Discount Factor $\gamma$ | 0.995 |
| GAE $\lambda$ | 0.95 |
| PPO Clip Value $\epsilon$ | 0.2 |
| Gradient Clip Value | 0.5 |
| Starting Learning Rate | 3e-4 |
| Number of Iterations per Update | 10 |
| Learning Rate Scheduler | Decay |
| Number of Processes | 8; 16 |
| Batch Size | 2048; 100 |
| Number of Timesteps | 8e6; 3e5 |
| Base Hidden Size $D$ | 128; 64 |

Table 2: RL Hyperparameters

For experiments with pretrained policy weights, we initialize the learned logstd to -1.0. For all models on the locomotion tasks, we perform a secondary search over the batch size in [256, 512, 1024, 2048] until performance decays. We find that MeMo works best with smaller batch sizes: 256 on the 12 leg centipede, 512 on the 10 leg worm, and 1024 on the 10 leg hybrid. The modular architecture also sees an improvement when using 1024 for all three robots, whereas the default batch size of 2048 works best for the MLP architecture. NerveNet-Conv and NerveNet-Snowflake improve with a batch size of 1024 on the 12 leg centipede and 10 leg worm. We do not see a significant improvement when decreasing the batch size for MetaMorph. Due to the higher variance in reward for grasping tasks, we keep the same batch size for the transfer experiments.

For the modular architectures, we parameterize each joint network within the modules with a 2 layer MLP with 32 hidden units per layer. For imitation learning, we use a batch size of 1024 and tune the learning rate in [7e-4, 1e-3, 2e-3, 4e-3, 7e-3]. All models are trained for 175 iterations of DAgger. We sample 500 trajectories from the expert controllers of the 6 leg centipede, 6 leg worm, and 6 leg hybrid and 250 trajectories from the controller of the 4 finger claw as the validation sets. We make sure that the architectures pretrained with imitation learning achieve a comparable average reward as the expert controller when a number of trajectories are sampled from them.

For NerveNet, the input network is a single layer with size $D$ followed by a Tanh nonlinearity. We have a separate output network for each joint, and each output network is a 2 layer MLP with 32 units per layer. Table 3 summarizes the hyperparameter search that we perform for NerveNet. We perform grid search over the number of layers and the size of the messages passed by the propagation network. We choose the smallest architecture size that achieves a similar average reward as MeMo. Adding a skip connection from the root to all joints improves NerveNet's validation score in imitation learning and enables the use of smaller architectures that do not overfit as easily.

| Parameters | Value Tried |
|---|---|
| Number of Layers | 2, 3, 4 |
| Message Size | 32, 64, 128 |
| Skip Connection | Yes, No |

Table 3: NerveNet Hyperarameter Search

For MetaMorph, we tune the number of attention layers in [2, 3, 4] and otherwise use the same architecture hyperparameters as [3], listed in Table 4. For a fair comparison, we use a MLP critic with the Transformer policy during RL rather than a Transformer critic, as the critic is trained from scratch during transfer. We find that positional encoding improves imitation learning pretraining, enabling the use of smaller architectures. For each token that corresponds to a joint, we include the one-hot encoding of the joint type in the observations.

| Parameters | Value |
|---|---|
| Number of Attention Heads | 1 |
| Embedding Dimension | 128 |
| Feedforward Dimension | 1024 |
| Nonlinearity | ReLU |
| Dropout | 0.1 |

Table 4: Transformer Hyperparameters

## A.7    State Space Description

We keep a running mean and variance to normalize the state space. Relative positions / orientations are relative to a joint in the same module as a given joint. For grasping, we use relative joint orientations as global joint orientations depend significantly on how high the claw is lifted. Table 5 and 6 detail the observation space in locomotion and grasping. For locomotion, "base" refers to the forwardmost wide body segment of the robot. As the joints are hinge joints, they only have one degree of freedom. The token type refers to the observation processing for MetaMorph – each input sequence consists of a single "global" token with the corresponding global observations for the robot concatenated with zero padding for the local observations, and the rest of the tokens are "joint" tokens with zero padding for the global observations concatenated with the local observations of the corresponding joint. We choose to integrate global information at the encoder level rather than the decoder level as our global features are low-dimensional: only 16 dimensions at most. The original MetaMorph architecture considers exteroceptive features from camera or depth sensors as global, which are much higher dimensional and are concatenated at the decoder level to prevent the dilution of local proprioceptive information.

## A.8    Computing Infrastructure

We run experiments on 2 different machines with AMD Ryzen Threadripper PRO 3995WX processors and NVIDIA RTX A6000 GPUs. Both machines have 64 CPU cores and 128 threads. The main cost of running the agent in both the RoboGrammar and the DiffRedMax environments is the cost of simulation, which is CPU-intensive. For the MLP-based architectures, we only use CPU cores for computing rollouts in parallel environments via vectorization and backpropagating the policy gradient. For NerveNet, in the locomotion domain, we find it helpful to vectorize environments while performing backpropagation with a GPU. For example, the RL stage of MeMo on the 6 leg centipede takes less than a day to complete, whereas training a 3 layer NerveNet-Conv with the same number of processes and batch size requires 3-4 days without a GPU. We note that our resources are shared, and the wallclock time varies depending on the other processes running on the same server.

## A.9    Additional environment details

We provide more details on the RoboGrammar tasks. On all locomotion tasks, the maximum episode length is 128. Full details of the environments can be found in the RoboGrammar codebase [16].

Table 5: Locomotion Observation Space

| Controller Type | Node Type | Token Type | Observation Type | Axis |
|---|---|---|---|---|
| boss | root | global | base position | y |
| | | | base velocity | x |
| | | | base velocity | y |
| | | | base velocity | z |
| | | | base angular velocity | x |
| | | | base angular velocity | y |
| | | | base angular velocity | z |
| | | | base orientation | x |
| | | | base orientation | y |
| | | | base orientation | z |
| boss, module | joint | joint | joint position | - |
| | | | joint velocity | - |
| | | | joint orientation | x |
| | | | joint orientation | y |
| | | | joint orientation | z |
| | | | joint relative position | x |
| | | | joint relative position | y |
| | | | joint relative position | z |

Table 6: Grasping Observation Space

| Controller Type | Node Type | Token Type | Observation Type | Axis |
|---|---|---|---|---|
| boss | root | global | relative fingertip position to object | x |
| | | | relative fingertip position to object | y |
| | | | relative fingertip position to object | z |
| boss, module | joint | joint | joint position | - |
| | | | joint velocity | - |
| | | | joint relative orientation | x |
| | | | joint relative orientation | y |
| | | | joint relative orientation | z |
| | | | joint relative position | x |
| | | | joint relative position | y |
| | | | joint relative position | z |

- Frozen Terrain: A flat surface with a friction coefficient of 0.05.
- Ridged Terrain: Ridges are placed an average of one meter apart across the width of the terrain.
- Gap Terrain: A series of platforms separated by gaps.
- Stepped Terrain: A series of steps with varying height, resembling a flight of stairs.

For all RoboGrammar locomotion environments, the reward at timestep $\tau$ is the sum of the rewards at each sub-step $t$. The training reward function at substep $t$ is

$$R(s_t, a_t) = V_x + 0.1(e_x^{\text{body}} \cdot e_x^{\text{world}} + e_y^{\text{body}} \cdot e_y^{\text{world}}) - 0.7\|a_t\|^2/N \tag{19}$$

where $N$ is the dimension of the action vector, and each dimension is normalized to [-1, 1]. The first two terms encourage high velocity in the $x$-direction and maintaining the robot's initial orientation respectively. The last term is a regularization penalty to reduce the variance across different runs. The reported reward curves do not include the regularization penalty.

For grasping, the goal is to grasp an object and lift it as high as possible and the maximum episode length is 50. As in prior work [6], we follow the convention of controlling the actuators with relative positions rather than absolute positions. The reward at timestep $\tau$ is the sum of the rewards at each sub-step $t$. The full set of parameters used to construct the DiffRedMax simulation will be released with our source code. Below is the reward function used, where object$_z$ refers to the object's z-coordinate and avg_fingertip_dist is the mean distance of the claw's fingertips to the object's surface. We approximate the cube's surface with the surface of the largest sphere that fits in the cube.

all_fingers_in_contact checks whether or not all fingers of the claw is within a small distance from the surface of the object.

$$R(s_t, a_t) = \begin{cases} 10 \cdot \text{object}_z - 0.1 \cdot \text{avg\_fingertip\_dist} & \text{all\_fingers\_in\_contact} \\ -0.1 \cdot \text{avg\_fingertip\_dist} & \text{!all\_fingers\_in\_contact} \end{cases}$$

The penalty on avg_fingertip_dist encourages the fingers to grasp the object. We only include the reward term on $\text{object}_z$ when all_fingers_in_contact is satisfied in order to prevent the claw from throwing the object.

### A.10    Sources

We use the PPO implementation provided in `https://github.com/ikostrikov/pytorch-a2c-ppo-acktr-gail` (MIT License). Our NerveNet implementation is adapted from a PyTorch version of the original NerveNet codebase: `https://github.com/HannesStark/gnn-reinforcement-learning`. We adapt our MetaMorph implementation from the official codebase: `https://github.com/agrimgupta92/metamorph/tree/main`. We use the official RoboGrammar [16] (MIT License) and DiffRedMax [17] (MIT License) simulators.

### A.11    Extended Related Works

**Noise Injection.** One line of work focuses on explaining the generalization benefits induced by noise injection by deriving explicit regularization terms. [25] studies Gaussian noise injected into network activations at each layer and derive an explicit regularization term by marginalizing out the noise. [26] analyzes the effect of Gaussian noise injection to the training data and find that the effect is equal to weighted ridge regularization as the number of noise injections approaches infinity. [27] study injecting noise into RNN hidden states and identify an explicit regularizer for small noise variances.

**Hierarchical and Multi-Task RL.** Our proposed modular architecture bears similarity to those used in hierarchical RL [28, 14]. However, a key difference is that our architecture is hierarchical with respect to the morphology of the robot, not the temporal structure of the task. To train robots that perform a diverse set of skills and generalize to new tasks, prior work leverages the shared structure of tasks, such as through graph representations [29, 30] that represent task compositionality, or through language representations [31, 32]. While many works in MTRL focus on a single morphology, recent efforts [33] have proposed representing both morphology and task in a single graph, enabling architectures trained on this unified IO representation to transfer to unseen morphologies and tasks.

**Multi-Robot Coordination.** Past works in multi-robot coordination bear similarity to our work in either the modularity of the architecture or the learning of a higher-level coordination mechanism, analogous to our boss controller, between different agents. In particular [34] uses a modular architecture, in which a higher-level meta-policy coordinates various skills with a behavior embedding. [35] proposes to learn a useful latent action space for coordinating a multiagent system via an information bottleneck. The information bottleneck helps in learning a latent action space from the full set of observations that is useful in coordinating decentralized agents at inference time.

### A.12    Additional Experiments

In Fig. 12, we test the zero-shot generalization of the pretrained NerveNet-Conv baseline by fixing all of its weights and only training the learned standard deviation when transferring from the 6 to the 12 leg centipede. Its poor performance demonstrates the difficulty of the transfer task, in spite of the physical similarities between the 6 and the 12 leg centipede.

In addition, we demonstrate the capability of MeMo is scaling to much higher dimensionalities by transferring from a 6 to a 24 leg centipede (Fig. 13) on Frozen Terrain. MeMo significantly outperforms RL (Modular), the strongest baseline on the 6 to 12 leg centipede transfer task.

In Fig. 14, we show that our framework has the potential to correct the dynamics of the system during transfer, specifically when some joints in the transfer robot fail to perform as expected even when given the correct control signal. In the 6 to 12 leg centipede transfer, we randomly select 7 out of 70 joints in the 12 leg centipede to be uncontrollable (for each seed of the experiment, a different subset

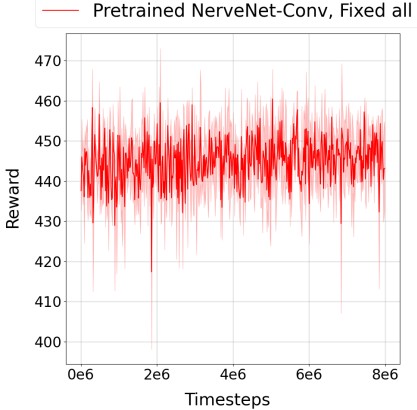

Figure 12: **Fixed NerveNet-Conv on 6 to 12 Leg Centipede Transfer.** All weights of the NerveNet-Conv baseline, pretrained on the 6 leg centipede, are fixed during transfer to the 12 leg centipede on the Frozen Terrain, resulting in suboptimal performance.

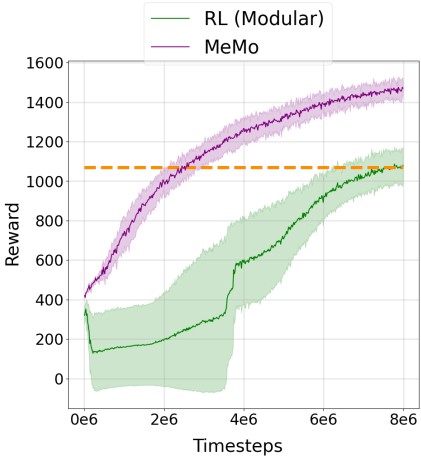

Figure 13: **6 Leg to 24 Leg Centipede Transfer Results.** The dashed orange line shows that the final performance of the closest baseline is achieved by MeMo in less than half of the total number of timesteps.

of uncontrollable joints is sampled, and we have 3 random seeds). For the uncontrollable joints, we pass in a small random noise instead of the controller's output to the simulator. MeMo significantly outperforms the RL (Modular) baseline, achieving its final reward in less than half of the timesteps.

In Fig. 15, we test the transfer capabilities of MeMo to morphologies smaller than the starting structure. Specifically, we transfer modules from a 6 leg to a 4 leg centipede on the Frozen Terrain and from a 4 finger to a 3 finger claw grasping a cube. MeMo achieves improved training efficiency to policies trained from scratch and performs similarly to the strongest baseline in each domain.

In Fig. 16, we compare the singular value distributions of MeMo to the ablations described in Section 5.3 and MeMo with a lower value of $\sigma$. In Fig. 17, we run MeMo on the 6 to 12 leg centipede transfer where either the boss controller or the modules are 4 layer MLPs instead of 2. Both of these variants perform similarly to the original architecture.

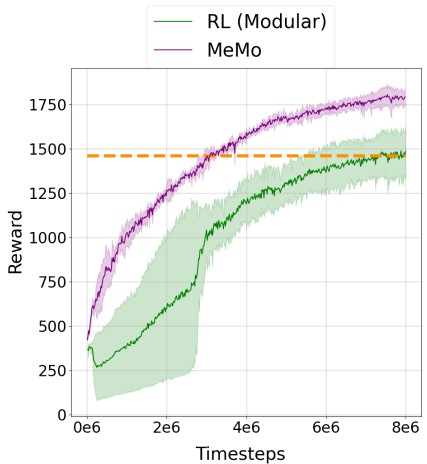

Figure 14: **6 Leg to Broken 12 Leg Centipede Transfer Results.** MeMo achieves the final performance of the closest baseline in less than half of the total number of timesteps.

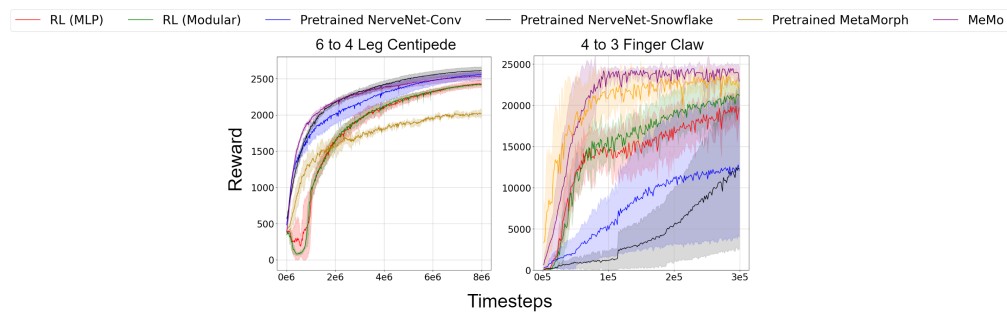

Figure 15: **Complex to Small Structure Transfer Results. Left:** 6 leg centipede to 4 leg centipede transfer on the Frozen Terrain. **Right:** 4 finger claw to 5 finger claw transfer on grasping a cube. On both transfer tasks, MeMo achieves comparable performance to the performance of the strongest baseline.

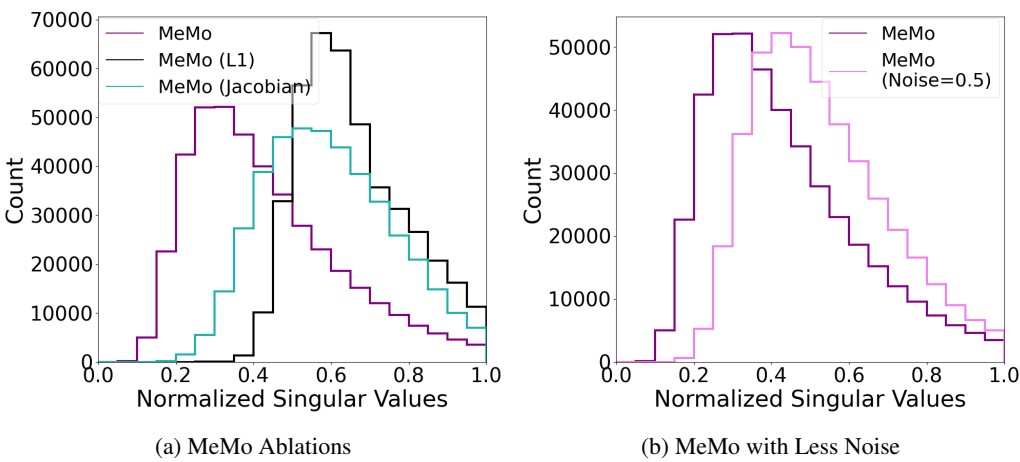

(a) MeMo Ablations

(b) MeMo with Less Noise

Figure 16: **Additional Singular Value Distributions. Left:** For various ablations of MeMo, we plot the normalized singular values of Jacobian matrices computed over an expert's trajectories. With noise injection, the mass of the distribution is much closer to 0. **Right:** With injected noise sampled from a Gaussian distribution with standard deviation 0.5 instead of 1.0, the mass of the distribution is closer to 1.

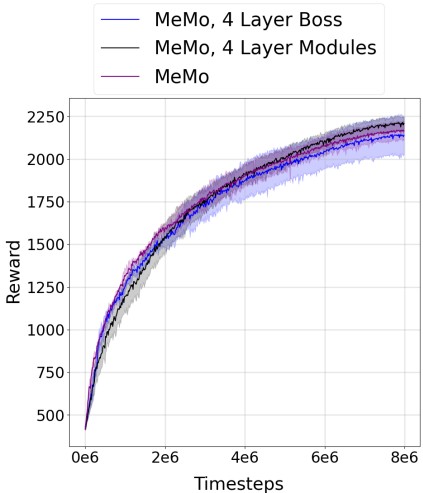

Figure 17: **Architecture Variants of MeMo on 6 to 12 Leg Centipede Transfer:** We run experiments where either the size of the boss controller or the size of the modules is increased from 2 to 4 layers. Both of these variants achieve comparable performance to the original architecture with 2 layer MLPs.

## A.13    Additional Figures

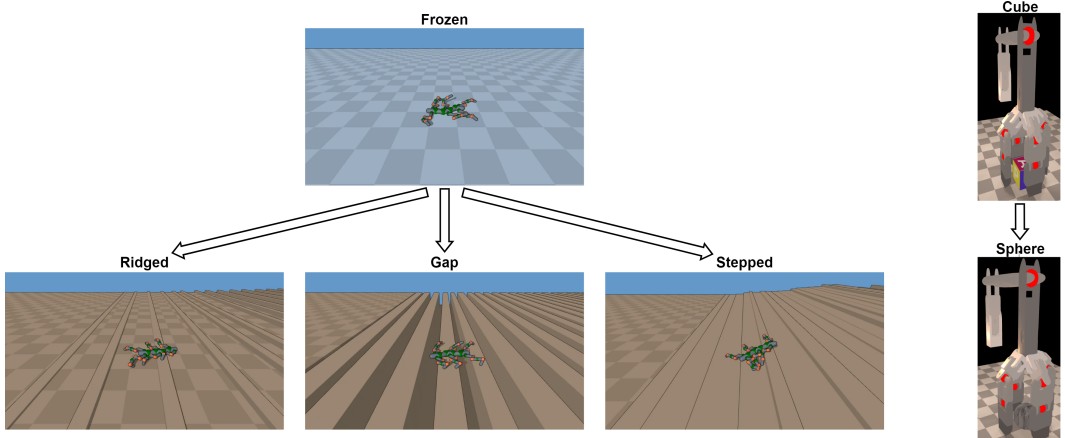

Figure 18: **Task transfer. Left:** The training task is a 6 leg centipede locomoting over the Frozen Terrain. The goal is to transfer policy weights to Ridged, Gap, and Stepped Terrains, all of which require the robot to overcome obstacles unseen in the Frozen Terrain. **Right:** In the grasping domain, the training task is a 4 finger claw lifting a cube, and the testing task is the same claw lifting a sphere. A sphere is naturally a harder object to grasp due to its curved surface.

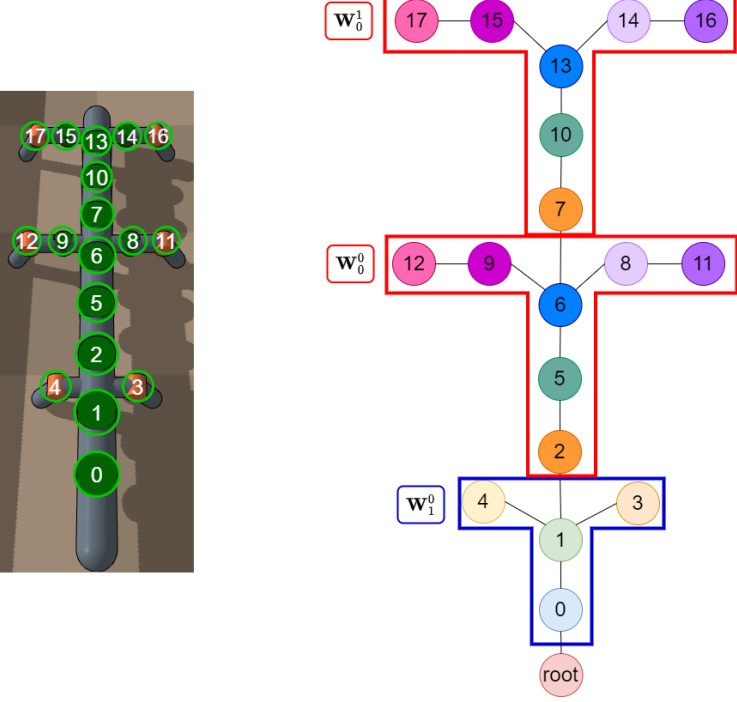

Figure 19: **Graph Structure and Modules of the 6 Leg Worm. Left:** Rendered robot, with joints labeled numerically and circled. **Right:** Corresponding graph structure with joints as nodes and links as edges. The joints circled in red can be thought of the "head" while the joints circled in blue form the "body" modules.

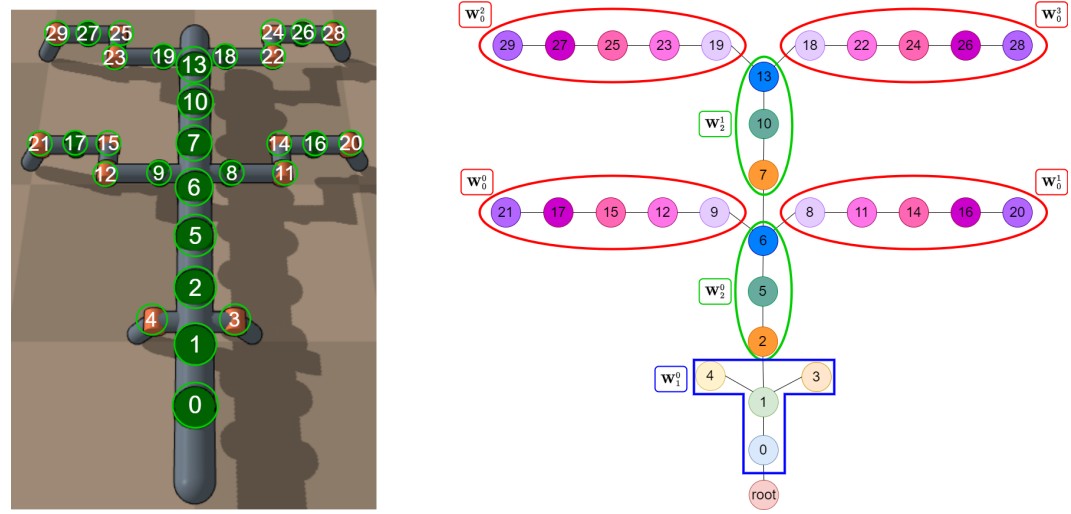

Figure 20: **Graph Structure and Modules of the 6 Leg Hybrid. Left:** Rendered robot, with joints labeled numerically and circled. **Right:** Corresponding graph structure with joints as nodes and links as edges. The joints circled in red belong to the "leg" modules, those circled in green belong to "body" modules, and those circled in blue belong to the "head" module.

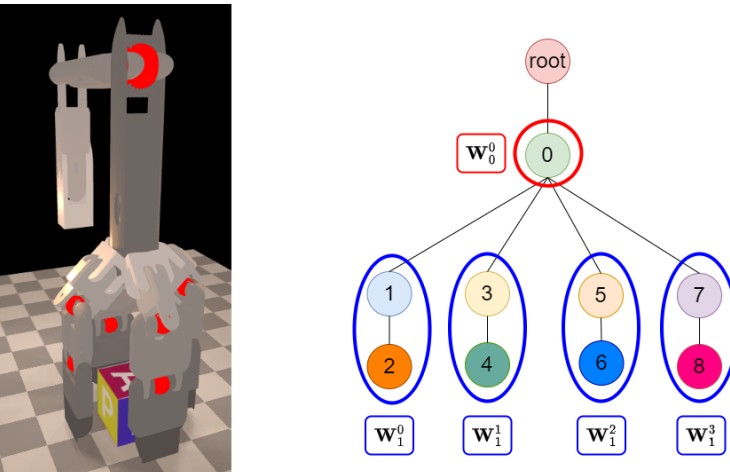

Figure 21: **Graph Structure and Modules of the 4 Finger Claw. Left:** Rendered robot, with joints denoted by red spheres. **Right:** Corresponding graph structure with joints as nodes and links as edges. Each pair of finger joints belongs in its own module, and the arm joint belongs in a separate module.

