# OpenReview forum: "MeMo: Meaningful, Modular Controllers via Noise Injection"
_NeurIPS.cc/2024/Conference — NeurIPS 2024 poster_

### Official Review · Reviewer_bS7d · 2024-06-24

**Soundness:** 3
**Presentation:** 2
**Contribution:** 3
**Rating:** 5
**Confidence:** 4

**Summary:**

This manuscript presents a hierarchical controller for reinforcement learning-based control. In particular, given a robot that can be decoupled into a high-level controller and a few low level (joint level) controllers, the two modules are learned jointly via a behavior cloning objective. The proposed method further makes the high-level controller robust to noise from the output of the low-level controllers by adding noise to their outputs. The model experiments on a few morphologies (6/12-leg centipede, 6/10-leg worm, etc) to demonstrate its effectiveness over vanilla RL and NerveNet.

**Strengths:**

1. Presents a hierarchical controller that enables sample efficient policy learning under morphological changes
2. Prior methods require training on many morphologies, then generalize to an unseen morphology. The presented method is trained on one morphology and then transferred to a new morphology.
3. Results suggest that the hierarchical formulation provides improved sample efficiency over baseline methods

**Weaknesses:**

1. Opposing strength 2, especially in simulation, many existing works randomize over many morphologies to enable efficient transfer between them or to an unseen morphology. It is unsure how this method can be trained on a dataset that contains more than one morphology.
2. For the experiments, the reviewer is uncertain how significant is the morphology change in affecting the dynamics of the system.

**Questions:**

1. Can the expert policy on 6-leg centipede, and 6-leg worm be executed on 12-leg centipede and the 10-leg worm? The same question applies to the manipulation setting (4-finger to 5-finger claw).
2. How effective is the method at correcting the dynamics of the system? I.e. after pre-training on the 6-leg centipede, if two or more of the pre-trained joints failed (i.e. uncontrollable), how fast can it adapt to the new setting?

**Limitations:**

Present in the manuscript.

---

> ### Author Rebuttal · Authors · 2024-08-05
>
> We thank the reviewer for the review. Below we address the reviewer’s concerns and questions.
>
> **Under "Weaknesses":**
>
> > Opposing strength 2, especially in simulation, many existing works randomize over many morphologies to enable efficient transfer between them or to an unseen morphology. It is unsure how this method can be trained on a dataset that contains more than one morphology.
>
> Our work is designed for the use case where the user does not have available to them a large number of morphologies. On the one hand, if you do have many morphologies, you can easily extend our approach to train the modules across the different morphologies and you may even get better modules; on the other hand, some of our baselines which are explicitly designed for such a use case may then outperform our approach.
>
>
> > For the experiments, the reviewer is uncertain how significant is the morphology change in affecting the dynamics of the system.
>
> One piece of evidence to demonstrate the difficulty of the structure transfer task is in Figure 12 (Appendix A.12), where we test the zero-shot generalization of NerveNet-Conv by fixing all weights on the 6 to 12 leg centipede transfer. If the change in morphology did not significantly affect the dynamics, we would expect the pretrained model to still perform relatively well. However, our experiments show that without finetuning, the NerveNet-Conv performance is much worse than before, showing that the coordination required at transfer time is significantly different from pretraining.
>
> **Under "Questions":**
> > Can the expert policy on 6-leg centipede, and 6-leg worm be executed on 12-leg centipede and the 10-leg worm? The same question applies to the manipulation setting (4-finger to 5-finger claw).
>
> In both locomotion and manipulation, our expert policy is a monolithic neural network, so due to the mismatch in the input and output space, it cannot be executed on our transfer morphologies. In addition, NerveNet [1] has MLP baselines in which they transfer weight matrices between hidden layers and they find that these baselines perform worse than their approach.
>
> [1] Wang et al. “NerveNet: Learning Structured Policy with Graph Neural Networks”
>
>
> > How effective is the method at correcting the dynamics of the system? I.e. after pre-training on the 6-leg centipede, if two or more of the pre-trained joints failed (i.e. uncontrollable), how fast can it adapt to the new setting?
>
> We are not entirely sure how to interpret the question, but if the reviewer means that there is a possibility that due to the narrow interface between the boss and modules, the boss is unable to have fine-grained control over the modules, we find that empirically the boss is still able to learn new dynamics during transfer time. For example, in the 6 to 12 leg centipede transfer, some of the legs in the 12 leg centipede have more limited range of motion than in the 6 leg to avoid collisions. Our results also show that our approach outperforms the baseline where we train the same modular architecture from scratch on the transfer morphology, even though the baseline has more flexibility to learn the appropriate dynamics.
>
> Alternatively, if the reviewer is referring to a scenario where during transfer time, some joints may fail to perform as expected even when given the correct control signal, we believe our framework would be able to adapt in this case as well. Our framework learns modules that represent control signals on a low-dimensional manifold with respect to the boss signal. Even though the manifold would change as a result of the pretrained joints behaving differently at transfer time, our pretrained modules still enable us to navigate along a low-dimensional manifold to infer the optimal coordination, enabling the Boss to be quickly adapted at transfer time.
>
> If neither response addressed the reviewer’s question, we are happy to provide more clarification.

---

> > ### Comment · Reviewer_bS7d · 2024-08-12
> >
> > Dear authors,
> >
> > Thanks for addressing my concerns regarding your paper and sorry for the belated review.
> >
> > > On the one hand, if you do have many morphologies, you can easily extend our approach to train the modules across the different morphologies and you may even get better modules
> >
> > Can you clarify how you may train the modules across different morphologies?
> >
> > > How effective is the method at correcting the dynamics of the system? I.e. after pre-training on the 6-leg centipede, if two or more of the pre-trained joints failed (i.e. uncontrollable), how fast can it adapt to the new setting?
> >
> > I am referring to the second setting mentioned in the response. It would be great if an experiment could be conducted in such a setting, so that we can see the trained policy is robust to change in dynamics..
> >
> > For my final rating, I will take into consideration the arguments which arise during the upcoming AC-Reviewers discussion phase. Overall, I lean towards increasing my score.
> >
> > Best regards!

---

> ### Author Response · Authors · 2024-08-13
> **Official Comment by Submission3440 Authors**
>
> > Can you clarify how you may train the modules across different morphologies?
>
> Consider an example where the training morphologies consist of the 4 leg centipede and 6 leg centipede, both of which are composed of “leg” and “body” modules, denoted as $\textbf{W}_0$ and $\textbf{W}_1$ respectively. Then the modular architecture for the 4 leg centipede is a composition of the boss $\textbf{B}_1$ and its set of modules $\mathcal{W}_1 =\\{\textbf{W}_0^0, …, \textbf{W}_0^3, \textbf{W}_1^0 \\}$ (4 leg modules and 1 body module). For each module, the subscript corresponds to module type and the superscript denotes different instances of the same module type, which share model parameters. Similarly, the 6 leg centipede is a composition of the boss $\textbf{B}_2$ and its set of modules $\mathcal{W}_2 = \\{\textbf{W}_0^4, …, \textbf{W}_0^9, \textbf{W}_1^1, \textbf{W}_1^2\\}$ (additional 6 leg modules and 2 body modules). While the boss controllers differ across various morphologies, the modules are shared. We can then train both architectures end-to-end with imitation learning while injecting independent noise vectors $\eta_1, \eta_2$ to the output of $\textbf{B}_1$, $\textbf{B}_2$ respectively at each batch.
>
> > I am referring to the second setting mentioned in the response. It would be great if an experiment could be conducted in such a setting, so that we can see the trained policy is robust to change in dynamics..
>
> Thanks for the clarification. We have run this experiment on the 6 leg to 12 leg centipede transfer task, where during transfer, we randomly select 7 out of 70 joints in the 12 leg centipede to be uncontrollable (for each seed of the experiment, a different subset of uncontrollable joints is sampled, and we have 3 random seeds). For the uncontrollable joints, we pass in a small random noise instead of the controller's output to the simulator. Below, we compare MeMo’s performance on this transfer task to RL (Modular), the strongest baseline from the 6 to 12 leg centipede transfer. MeMo significantly outperforms the baseline, achieving its final reward in less than half of the timesteps. This demonstrates the potential of our framework to adapt to unforeseen dynamics at transfer time.
>
> | |4e+6 timesteps | 8e+6 timesteps|
> | :---------------- | :------: | ----: |
> |RL (Modular)| 1203|1485 |
> |MeMo|**1591** |**1793** |
>
> Edit: Added table with final rewards.

---

### Official Review · Reviewer_tVBX · 2024-07-10

**Soundness:** 3
**Presentation:** 3
**Contribution:** 3
**Rating:** 6
**Confidence:** 2

**Summary:**

The MeMo framework presented in this paper proposes an innovative approach for enhancing the transferability of control systems across robots with varied morphologies by utilizing pre-trained, modular controllers. This method facilitates rapid adaptation to new robot designs by leveraging previously trained modules, thereby significantly improving training efficiency over conventional methods like graph neural networks and Transformers.

**Strengths:**

The presentation looks good. The results presented are intriguing and demonstrate potential.

**Weaknesses:**

N/A

**Questions:**

1. The comparative analysis raises some questions regarding the fairness of the evaluation. Specifically, were the control modules of the compared methods also retrained on your newly designed robots, or was it only the master controller that was retrained? If the latter is the case, this might lead to an unfair comparison since these pre-trained models have not been directly exposed to the newly designed robot configurations prior to testing.
2. The transition from simulation to real-world applications is not addressed. The paper would benefit from a discussion on the expected challenges when implementing these modules in actual robotic systems, such as dealing with hardware inconsistencies and environmental variations. For example, how does the framework handle discrepancies in the physical properties between different robot assemblies, such as variations in leg mechanics?

---

> ### Author Rebuttal · Authors · 2024-08-05
>
> We thank the reviewer for their feedback and are glad that they find our results intriguing. Below we address the reviewer’s questions.
>
>
> **Under "Questions":**
> > The comparative analysis raises some questions regarding the fairness of the evaluation. Specifically, were the control modules of the compared methods also retrained on your newly designed robots, or was it only the master controller that was retrained? If the latter is the case, this might lead to an unfair comparison since these pre-trained models have not been directly exposed to the newly designed robot configurations prior to testing.
>
> For MetaMorph, we finetune the Transformer and the joint modules, as in the original work. For the NerveNet baselines, while parts of the network are fixed, prior work has found that such weight fixing actually improves transfer performance [1]. In the case of NerveNet-Conv, we experiment with both fixing the joint modules and finetuning the entire architecture and find that performance improves by fixing the joint modules.
>
> [1] Blake et al. “Snowflake: Scaling GNNs to High-Dimensional Continuous Control via Parameter Freezing”
>
> > The transition from simulation to real-world applications is not addressed. The paper would benefit from a discussion on the expected challenges when implementing these modules in actual robotic systems, such as dealing with hardware inconsistencies and environmental variations. For example, how does the framework handle discrepancies in the physical properties between different robot assemblies, such as variations in leg mechanics?
>
> We agree that sim-to-real transfer is an important problem; however, the goal of learning controllers that are robust to variations outside of simulation is orthogonal to our problem of learning modules that generalize to different morphologies. We note that our baselines, NerveNet and MetaMorph, also perform their experiments only in simulation. In our revised paper, we will extend our discussion of sim-to-real transfer as an important line of future work. At a high level, we would expect our framework to face the same challenges in adapting to the real world as standard RL policies, including discrepancies in simulated physics vs real dynamics and, as the reviewer mentioned, hardware inconsistencies and environmental variations.

---

> ### Comment · Reviewer_tVBX · 2024-08-12
> **Thank authors for detailed explanations**
>
> Thank authors for detailed explanations. I don't have any other questions and am happy to raise my points to 6.

---

> > ### Author Response · Authors · 2024-08-12
> > **Thank you for your response**
> >
> > Thank you very much -- we're happy we were able to address your questions!

---

### Official Review · Reviewer_5WVH · 2024-07-13

**Soundness:** 3
**Presentation:** 3
**Contribution:** 2
**Rating:** 6
**Confidence:** 4

**Summary:**

This paper introduces a new framework designed to create modular controllers allowing for quicker adaptation of control strategies when building new robots with similar methodology. The MeMo framework employs a novel modularity objective optimized alongside standard behavior cloning loss through noise injection. Experiment results in locomotion and grasping environments, ranging from simple to complex robot morphologies, demonstrate that MeMo improves training efficiency compared to multiple baselines. Additionally, MeMo’s modular approach enhances both structure and task transfer capabilities.

**Strengths:**

* The proposed method utilizes noise injection to build meaningful modular controllers that suit the application scenario as described: transfer the controller to a robot with similar but not identical morphology. The proposed method is easy to understand and should be easy to implement
* This work provides reasonable grounding of the proposed method in Sec 3.1
* The experiment shows the proposed method outperforms all baselines in locomotion and manipulation tasks as shown in Figure 6/7, in terms of better sample efficiency and better final performance in some environments.
* This work provides a decent ablation study on the noise injection to validate the component's importance. including replacing the noise injection with different forms of regulation.
* The paper is well-written and easy to follow, a reasonable amount of technical details are included in the method and supplementary materials.
* The analysis of the learned module in section 5.4 and Figure 15 looks interesting.

**Weaknesses:**

* The tasks evaluated are relatively simple, only basic locomotion tasks and quite simple manipulation tasks are considered.
* All tasks involve only a rigid body, which could be simulated much faster with GPU with the recent simulator, which should deliver a much shorter training time compared to the reported time in the appendix. It’s a bit questionable whether the proposed method is necessary if training a new policy from scratch takes a short period.

**Questions:**

No specific questions to add.

**Limitations:**

This work has provided reasonable discussion over the limitation and no potential negative social impact of the work needs to be considered.

---

> ### Author Rebuttal · Authors · 2024-08-05
>
> We thank the reviewer for the review and are glad that they find our method easy to understand and empirical evaluation thorough. Below we address the reviewer’s concerns and questions.
>
> **Under "Weaknesses":**
>
> > The tasks evaluated are relatively simple, only basic locomotion tasks and quite simple manipulation tasks are considered.
>
> Although relatively simple, our tasks enable us to extensively evaluate the structure transfer capabilities of our framework, which is the primary focus of our work.
>
> > All tasks involve only a rigid body, which could be simulated much faster with GPU with the recent simulator, which should deliver a much shorter training time compared to the reported time in the appendix. It’s a bit questionable whether the proposed method is necessary if training a new policy from scratch takes a short period.
>
> While we agree that the training time can be faster with recent simulators, the final reward of policies trained from scratch are below that of MeMo in our structure transfer experiments, an issue that cannot be fixed by faster simulation. These advances in simulation would also benefit MeMo, allowing the Boss controller to be retrained more quickly.

---

> > ### Comment · Reviewer_5WVH · 2024-08-12
> >
> > Thank the authors for addressing my concerns, I would like to keep my original evaluation.

---

### Official Review · Reviewer_Hbiz · 2024-07-13

**Soundness:** 3
**Presentation:** 3
**Contribution:** 3
**Rating:** 6
**Confidence:** 3

**Summary:**

This paper presents a method for learning modular controller that can be transferred and adapted to different morphology of robots and tasks. The high level idea is to decompose control of each motor through distilling a learned hierarchical RL policy and then uses these primitive policies as building blocks to additional motors added.

**Strengths:**

This paper proposes a way to transfer low-level control policies to similar but different morphology of robots with a learned structure. The problem is novel and interestingly challenging. The proposed method is conceptually intuitive. Experiments in simulation demonstrate that the proposed algorithm can enable policy transfer across different robots and tasks.

**Weaknesses:**

The proposed method only trains the master policy at adaptation stage, which assumes the change of morphology does not induce change in dynamics of the system such that drastically different low-level policies will be needed to solve the task.

**Questions:**

Will this method generalize to multi-agent RL setting where the modular policy might be more complex than controlling 1-DoF motor?

**Limitations:**

see weakness

---

> ### Author Rebuttal · Authors · 2024-08-05
>
> We thank the reviewer for the review and are glad that they find our problem novel and our proposed framework intuitive. Below we address the reviewer’s concerns and questions.
>
> **Under "Weaknesses":**
>
> > The proposed method only trains the master policy at adaptation stage, which assumes the change of morphology does not induce change in dynamics of the system such that drastically different low-level policies will be needed to solve the task.
>
> This is correct, and we will be happy to make this clearer in the final version of the paper; we expect our method to be most effective in situations like the ones in our experiments, where the new morphology is performing a similar task to the old morphology, and therefore there is a reasonable expectation that the low-level controllers learned with the initial morphology will be useful in the new morphology. In contrast, a situation where our technique may not be expected to work as well would be one where a module is used in a drastically different context; for example, pretraining a module as a leg for a crawling robot and later using it as a finger in a hand, which may require a very different range of motions even if it reuses the same parts. That said, our task transfer experiments show that the low-level controllers learned through our technique can be useful when transferring to a different but similar task. Even when the task is sufficiently different as to require different local controllers, e.g. the modules used to walk over a flat terrain are finetuned when transferring to a terrain with steps, using the pretrained controllers as a starting point can reduce the training cost.
>
>
> **Under "Questions":**
>
> > Will this method generalize to multi-agent RL setting where the modular policy might be more complex than controlling 1-DoF motor?
>
> To clarify, each local module does control subassemblies with more than one DoF -- for instance, the leg module of the centipede controls 5 joints. However, we agree that multi-agent RL would likely require more complex low-level controllers. At an abstract level, the problem of multi-agent RL is quite similar to what we address in our work -- the multi-agent policy can be decomposed into a higher-level controller that coordinates lower-level controllers corresponding to the individual agents. So while we have not experimented in this setting, we envision that our method could be applied to learn an appropriate division of labor between the boss that coordinates multiple agents and the more complex modules.

---

> ### Comment · Reviewer_Hbiz · 2024-08-12
> **thanks for clarification**
>
> I have read the authors' response and appreciate the clarification. It would be interesting to see extending this method to multi-agent RL settings!

---

### Decision · Program_Chairs · 2024-09-25

**Decision:**

Accept (poster)

**Comment:**

The paper tackles an interesting and challenging problem. The reviewers appreciated the replies, additional details, and additional experiments. Overall the proposed method is an important first step, however there is also quite some way to go to make it more general (e.g. for more complex systems, multi-agent setting). On the flip-side focusing on simpler systems allows for an in-depth analysis.
After the discussion all reviewers are in favor of accepting this paper.